# Progressive changes in coral reef communities with increasing ocean acidification
Sam H. C. Noonan [1] ✉, Chico Birrell[2], Rebecca Fisher [3,4] & Katharina E. Fabricius [1]

Ocean acidification from increasing atmospheric $CO_2$ is progressively affecting seawater chemistry, but predicting ongoing and near-future consequences for marine ecosystems is challenging without empirical field data. Here we quantify tropical coral reef benthic communities at 37 stations with varying exposure to submarine volcanic $CO_2$ seeping, and determine the aragonite saturation state ($\Omega_{Ar}$) where significant changes occur in situ. With declining $\Omega_{Ar}$, reef communities displayed progressive retractions of most reef-building taxa and a proliferation in the biomass and cover of non-calcareous brown and red algae, without clear tipping points. The percent cover of all complex habitat-forming corals, crustose coralline algae (CCA) and articulate coralline Rhodophyta declined by over 50% as $\Omega_{Ar}$ levels declined from present-day to 2, and importantly, the cover of some of these groups was already significantly altered at an $\Omega_{Ar}$ of 3.2. The diversity of adult and juvenile coral also rapidly declined. We further quantitatively predict coral reef community metrics for the year 2100 for a range of emissions scenarios, especially shared socio-economic pathways SSP2-4.5 and SSP3-7.0. The response curves show that due to ocean acidification alone, reef states will directly depend on $CO_2$ emissions, with higher emissions causing larger deviations from the reefs of today.

Human activities are increasing atmospheric carbon dioxide ($CO_2$) concentrations at a rate unprecedented for at least the last 66 million years[1,2]. Approximately one quarter of this $CO_2$ is absorbed by the surface waters of the world's oceans, increasing the partial pressure of $CO_2$ ($pCO_2$), lowering pH and causing the chemical changes known as ocean acidification[3]. Oceanic pH is now lower than it has been for over 800,000 years, and this will likely be irreversible for millennia[4,5]. Ocean acidification is also lowering the saturation state of aragonite ($\Omega_{Ar}$), with levels declining in tropical waters at a rate of ~0.1 units per decade[6,7]. Importantly, ocean acidification is not a future-only problem, and while the ongoing chemical changes are increasingly well documented[7–9], the present-day and near-future responses of marine communities remain less well understood.

Most ocean acidification studies compare biological responses between present-day control conditions and one or few levels of increased $CO_2$, typically under artificial experimental conditions[10]. While these studies have formed the basis of our understanding of the effects of ocean acidification, they are unable to investigate any continual changes in the responses[11]. Biological responses to a changing environment may be gradual and smooth, or abrupt when thresholds are present and a sharp transition or change in the slope of the response curve occurs over a smaller environmental change[11–14]. With few treatment levels it is not possible to identify the minimum increase in $CO_2$ that will result in a significant biological response[15], as this may be occurring at $CO_2$ levels below those typically used as experimental treatments[16]. Hence experimental outcomes and the interpretation of effects can depend on the number of experimental treatments, their concentrations and exposure times[10,14]. Furthermore, laboratory experiments can only approximate in situ environmental conditions and are largely void of species interactions, limiting their extrapolation to communities in natural environments. There is thus a need to study marine communities exposed to multiple levels of ocean acidification in situ, to better understand how this ongoing change is shaping communities today and in the near future.

Coral reefs are particularly vulnerable to ocean acidification because they are built by organisms with calcium carbonate skeletons ($CaCO_3$), and lowered pH and $\Omega_{Ar}$ can inhibit calcification and accelerate $CaCO_3$ dissolution[17]. Scleractinian corals have long been considered susceptible to these effects due to their aragonite skeletons[10,18,19], and the high-Mg-calcite skeletons of crustose coralline algae (CCA) and other calcareous algae have

[1]Australian Institute of Marine Science, Townsville, Australia. [2]General Organization for Conservation of Coral Reefs and Turtles in the Red Sea, Jeddah, Kingdom of Saudi Arabia. [3]Australian Institute of Marine Science, Perth, Australia. [4]Oceans Institute, University of Western Australia, Perth, Australia. ✉ e-mail: s.noonan@aims.gov.au

**Fig. 1 | Carbon chemistry at each of the 37 stations along the $CO_2$ gradient.** Measured values of mean $pH_{Total}$ ($pH_T$) and total alkalinity ($A_T$) (**a**), and $pH_T$ and calculated aragonite saturation state ($\Omega_{Ar}$) (**b**). Error bars are one standard error.

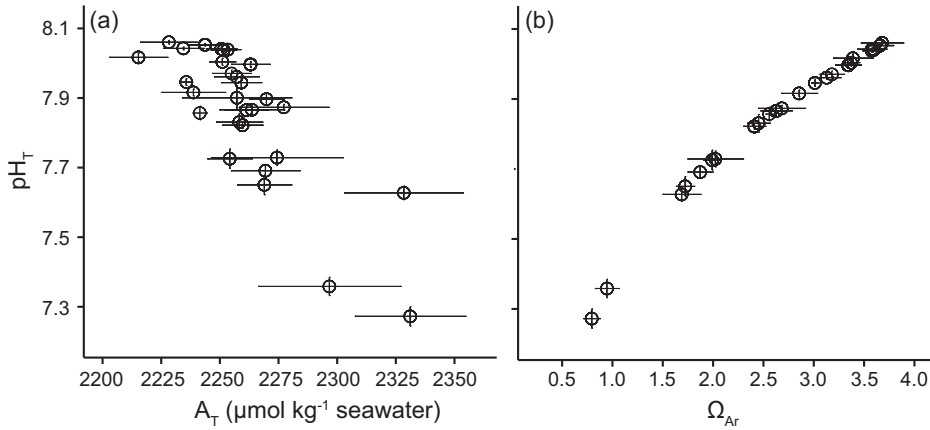

been repeatedly shown to be particularly sensitive[20,21]. The reef matrix itself and sediments are largely made of $CaCO_3$ and may be impacted sooner than many organisms as they have large surface-area to volume ratios and are not isolated from the adjacent seawater by tissue layers[22]. Ocean acidification also increases dissolved inorganic carbon ($C_T$) concentrations, which can stimulate photosynthesis in certain circumstances[23,24]. Some non-calcareous photosynthetic reef organisms are predicted to benefit from ocean acidification if their photosynthesis is stimulated and calcareous competitors are simultaneously constrained[25].

Investigations of the effects of ocean acidification on reef communities have been based on mesocosm experiments[26–28] and observations at naturally occurring high $CO_2$ analogues, e.g. volcanic $CO_2$ seep sites[29–32] or other oceanographic features affecting carbon chemistry[33–35]. While substantial variation exists between studies, they generally show consensus with the larger existing body of experimental work: that ocean acidification is likely to negatively affect many calcareous taxa (e.g. corals and CCA), while benefiting some non-calcareous groups (e.g. certain algae, sponges and seagrass). However, the magnitude and rate of change to reef communities under progressive ocean acidification remains largely unknown, including whether the changes will be linear or will occur at any abrupt tipping point[11].

In this study we compare coral reef benthic communities at 37 stations representing a range of long-term ocean acidification conditions around volcanic $CO_2$ seeps where strings of pure $CO_2$ gas bubbles emerge from the sea floor. Except for contrasting levels of $CO_2$ exposure, environmental conditions (e.g. levels of light, temperature, salinity and water flow) were similar across stations in this unique natural laboratory. We examined changes in the cover of common taxa, the density of young corals, and the biomass of different algal groups along the $CO_2$ gradient. For the first time, these response curves allowed us to quantify the minimum $CO_2$ increases that resulted in significant reef functional changes, and to project future changes to key coral reef benthos under different $CO_2$ emission scenarios.

## Results

### Carbon chemistry

The stations were distributed across a field of patches with contrasting seeping activity, extending from areas unaffected by volcanic $CO_2$ reflecting today's mean carbon chemistry into patches with dense bubble streams representing predicted mean future conditions (Fig. 1; Supplementary Table 1). Ambient stations unaffected by the volcanic $CO_2$ recorded total pH ($pH_T$) values averaging 8.02 ± 0.01 SE. Fifteen stations within the seep area were within 0.2 pH units of ambient values, and the remaining seven stations recorded a mean $pH_T$ below 7.8 (minimum station mean $pH_T$: 7.27 ± 0.03). At ambient stations, total alkalinity ($A_T$) averaged 2253 ± 1.16 μmol kg⁻¹ seawater, and $A_T$ increased along the gradient of seep exposure to 2331 μmol kg⁻¹ SW (Fig. 1). The calculated saturation state of aragonite ($\Omega_{Ar}$) was highly correlated with $pH_T$, but increased relative to $pH_T$ at $pH_T$ values below ~7.7 due to their elevated $A_T$, likely due to

increased $CaCO_3$ dissolution. $\Omega_{Ar}$ at stations unaffected by volcanic $CO_2$ averaged 3.57 ± 0.02 while stations exposed to volcanic $CO_2$ had $\Omega_{Ar}$ ranging from 3.39 to 0.79. Due to its significance for calcifying organisms and its high correlation to pH and other carbonate chemistry parameters, $\Omega_{Ar}$ was chosen as a proxy to characterise the ocean acidification level of each station. Temperature did not change along the carbon chemistry gradient (SeaFET logger data, GLM, p > 0.05), ranging from 27.5 – 28.8 °C across stations (Supplementary Table 1).

### Coral communities—benthic cover

A redundancy analysis indicated that changes in the reef communities across the stations were strongly related to $\Omega_{Ar}$ (PERMANOVA: $\Omega_{Ar\,(1,\,35)}$, $F = 2.594$, $p = 0.001$). In the ordination plot, most stations with the highest $\Omega_{Ar}$ clustered closely together, with positive RDA1 axis values that were associated with an increased percent cover of many calcareous taxa, including the important reef habitat-building hard corals *Acropora*, *Pocillopora*, *Seriatopora*, *Goniastrea* and *Fungia* spp., as well as articulate calcareous Rhodophyta and crustose coralline algae (CCA) (Fig. 2, Supplementary Table 3). Station benthic communities with mean $\Omega_{Ar}$ values < 3.0 typically recorded negative RDA1 values (Fig. 2, Supplementary Table 3). Seep site stations with lower $\Omega_{Ar}$ were typically associated with higher cover of noncalcareous taxa including brown and red macroalgae, sponges and the soft coral *Sarcophyton* spp., as well as the hard coral *Porites* (Fig. 2).

The in situ data documented continual changes in reef benthic communities in response to near-future ocean acidification (Fig. 3, Table 1). In general, abrupt thresholds were not detected, but rather we observed continual smooth changes along the $\Omega_{Ar}$ gradient. Changes were either linear or log-linear, with different slopes or susceptibilities between taxa. The percent cover of all scleractinian corals combined did not change along the $\Omega_{Ar}$ gradient and averaged 36.15 ± 1.59% across all stations (generalised linear models, Supplementary Table 2). This was largely driven by the cover of massive *Porites* spp., which was unaffected by $\Omega_{Ar}$. However, without massive *Porites* spp., the cover of all remaining hard corals combined declined compared to ambient values, by 20% at $\Omega_{Ar}$ 3.0, and by 35% at $\Omega_{Ar}$ 2.5 (Fig. 3, Table 1). The combined cover of structurally complex hard corals (i.e. branching, corymbose and tabulate growth forms), which are important as habitat for a large proportion of reef-associated organisms, was highly susceptible. Here the no-significant-effect concentration (NSEC), being the $\Omega_{Ar}$ value where the mean response first occurs outside the 99% confidence intervals of the response at ambient $\Omega_{Ar}$ values, indicated that complex coral cover became significantly lower than control values with a reduction in $\Omega_{Ar}$ of <0.4 units (NSEC $\Omega_{Ar} = 3.11$). This decline matched patterns in the overall complexity score for each quadrat (Table 1). The cover of the genera *Acropora*, *Seriatopora* and *Pocillopora* were all highly susceptible and reduced by at least 30% by $\Omega_{Ar}$ 3.0. The combined cover of non-complex hard corals (i.e. massive, submassive and encrusting growth forms) was

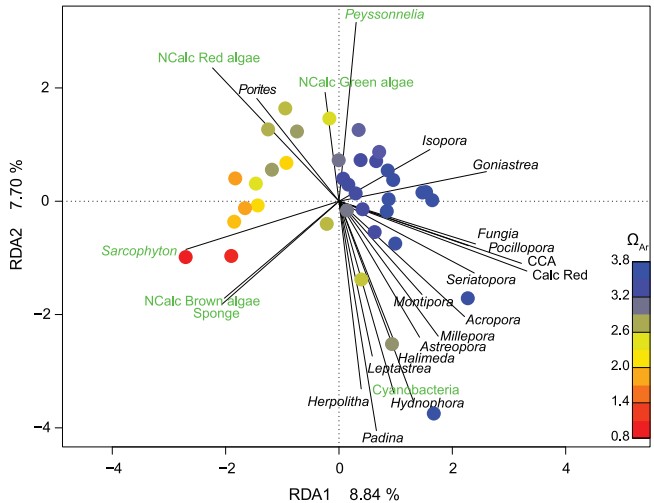

**Fig. 2 | Redundancy analysis ordination plot of the 37 benthic reef communities with contrasting aragonite saturation states ($\Omega_{Ar}$).** Points represent station means and their colours represent their station mean aragonite saturation state ($\Omega_{Ar}$). The top 50% of the most influential vectors are shown. Vector labels for noncalcareous taxa (NCalc) are written in green, for calcareous taxa in black. All vector labels and scores are listed in Supplementary Table 3.

unaffected by the $\Omega_{Ar}$ gradient, again driven by massive *Porites* cover. However, even some robust taxa such as *Goniastrea* spp., and the Merulinidae and Fungiidae declined along the $\Omega_{Ar}$ gradient (Table 1, Supplementary Fig. 2). While no clear threshold was seen, virtually no other corals besides massive *Porites* spp. were found at stations with a mean $\Omega_{Ar} < 2.0$ (Fig. 3). The other hard coral families recorded (Agariciidae, Diploastraeidae, Lobophyllidae and Euphylliidae) averaged <1% cover each and were not analysed individually. Soft coral cover was also low (1.28% ± 0.36) and statistically unaffected by $\Omega_{Ar}$, although practically no soft corals were found below $\Omega_{Ar}$ 2.5. Sponge cover increased along the $\Omega_{Ar}$ gradient by >150% of control values at $\Omega_{Ar}$ 2.5.

### Coral diversity and juvenile densities

Hard coral diversity, and hard and soft coral juvenile densities and diversity, all declined along the $\Omega_{Ar}$ gradient (Fig. 4, Supplementary Table 2). Adult and juvenile hard coral diversity were highly correlated (Pearson's correlation: $t = 7.91$, df = 31, $p < 0.001$), and both reduced significantly from ambient values as $\Omega_{Ar}$ declined by as little as 0.3 units (NSEC $\Omega_{Ar} = 3.20$). Adult soft coral diversity was low, averaging 0.04 ± 0.16 genera m$^{-2}$ across all sites, and was statistically similar along the $\Omega_{Ar}$ gradient. Soft coral juveniles were also sparse, but more sensitive to $\Omega_{Ar}$ declines than the hard corals. The diversity and density of soft coral juveniles declined by 50% between $\Omega_{Ar}$ 3.5 to 3.0, and none were found below $\Omega_{Ar} \sim 2.6$. This was approximately the same $\Omega_{Ar}$ value at which adult soft coral percent cover was reduced to zero.

### Macroalgal cover and biomass

The cover of algal communities also changed along the $\Omega_{Ar}$ gradient, with a general shift from calcareous to non-calcareous taxa from ambient $\Omega_{Ar}$ into the seeps (Fig. 3). The percent cover of all calcareous algae combined at control sites declined by 20% and 35% at $\Omega_{Ar}$ 3.0 and 2.5, however susceptibility differed between taxa. The abundant heavily calcified articulate Rhodophyta were the most affected, their cover was significantly reduced between ambient $\Omega_{Ar}$ and 3.25, and none were found at <2.4. Crustose coralline algae (CCA) were also highly sensitive, and cover declined by 40% between ambient and $\Omega_{Ar}$ 3.0. Both algal groups were virtually non-existent at $\Omega_{Ar} \leq 2.5$. In contrast, turf algae and the weakly calcified red algae *Peyssonnelia* spp. were unaffected by $\Omega_{Ar}$. Total non-calcareous macroalgal cover increased as $\Omega_{Ar}$ declined. This was due to non-calcareous brown and red algae, which similarly increased cover by 65% and 85% from ambient

values to $\Omega_{Ar}$ 2.5. *Halimeda* spp. and *Padina* spp. (the only calcareous green and brown algae recorded), and non-calcareous green algae each had <0.5% cover and were too low to analyse individually.

The biomass (dry weight) of all collected macroalgae combined was highest at $\Omega_{Ar}$ 3.5 and declined 15% and 30% by $\Omega_{Ar}$ 3.0 and $\Omega_{Ar}$ 2.5. This was due to high calcareous macroalgal biomass at controls, which declined by 25% at $\Omega_{Ar}$ 3.0, and by 50% at $\Omega_{Ar}$ 2.5 (Fig. 5). Calcareous algae made up 50% of total macroalgal biomass at the controls, but this ratio declined to <25% at $\Omega_{Ar}$ 2.5. Total non-calcareous macroalgal biomass was unaffected by $\Omega_{Ar}$, averaging 49.41 ± 6.08 g m$^{-2}$ across all stations. This was due to a replacement of high *Turbinaria* spp. at ambient $CO_2$ by *Melanamansia* spp. as $CO_2$ increased: at $\Omega_{Ar}$ 2.5 *Turbinaria* spp. were almost completely absent, while *Melanamansia* spp. more than doubled. Total non-calcareous algal biomass, without *Turbinaria* spp., greatly increased along the $\Omega_{Ar}$ gradient (Fig. 5).

## Discussion

Our empirical data from this unique natural laboratory demonstrate the drastic rates of change in coral reef benthic communities along a gradient of declining $\Omega_{Ar}$. At present-day $CO_2$, these coral reefs support a diverse community of scleractinian corals and calcareous algae. These are increasingly replaced by some non-calcareous algae, massive *Porites* spp. corals and sponges as $CO_2$ concentrations increase. High-$CO_2$ communities are also increasingly less diverse and less structurally complex. Our study uniquely served to quantify near-future changes in reef communities from ocean acidification, and to define the $\Omega_{Ar}$ values that result in a significant deviation from the reefs of today. The results add to a large body of work predicting fundamental changes to coral reef communities and functions by the end of the century, with the magnitude of change dependent on the atmospheric $CO_2$ emissions pathway realised. Alarmingly, these results also show that reef community changes are likely already occurring within the range of dissolved $CO_2$ levels observed on contemporary reefs[16,36].

The few studies with sufficient levels of $CO_2$ to investigate continual changes to reef communities under increasing ocean acidification have shown both linear and threshold responses. Laboratory experiments on individual organisms have shown mostly linear effects[11], while results of community-based studies or those in situ suggest tipping points may also occur[13,16]. For example, Smith et al.[16] and Kleypas et al.[37] both suggest that there will be a threshold in coral and algal community change between $\Omega_{Ar}$ 3.4 and 3.6, and $\Omega_{Ar}$ 3.0 has been suggested as the limit for global reef development[37,38]. We also previously deployed standardised settlement substrate along a gradient of increasing $CO_2$ at this and other seep sites in Papua New Guinea, and found newly recruited CCA communities were almost absent on these surfaces at mean $\Omega_{Ar} < 2.5$[13]. In the present study, we found linear or log-linear changes in most parameters measured along the $CO_2$ gradient, beginning immediately as $\Omega_{Ar}$ declined from ambient values, and although CCA and several complex coral taxa reached very low values at 2.5 $\Omega_{Ar}$, there was little evidence of a firm tipping point or threshold for the reef communities. Here communities began to significantly change from the present-day as $\Omega_{Ar}$ values declined by as little as 0.25 units. This immediate progressive response without thresholds highlights two important considerations. Firstly, any coral reef community thresholds in response to $\Omega_{Ar}$ declines may have already been exceeded, as global $\Omega_{Ar}$ values are now 0.5 units lower than at the advent of the Industrial Revolution[9,39]. Yet we are using present-day values as our baseline and cannot infer the shape of response curves at higher $\Omega_{Ar}$ values. Secondly, the lack of thresholds indicates that even small deviations in seawater $\Omega_{Ar}$ from present-day ambient values will continue to alter coral reef communities, and the magnitude of change that communities experience into the future will likely directly depend on the amount of $CO_2$ in the atmosphere.

We documented a range of contrasting susceptibilities between taxa along the $\Omega_{Ar}$ gradient. The NSEC values indicated habitat complexity was the most sensitive metric, followed by the cover of articulate calcareous red algae, the ratio of calcareous to total algal biomass, coral diversity and juvenile coral density, then the cover of several complex habitat-forming

**Fig. 3 | Changes in benthic cover of reef coral and algal communities in relation to aragonite saturation state ($\Omega_{Ar}$).** The plots show percent cover of the resilient massive *Porites* spp. coral (**a**), all hard corals excluding massive *Porites* spp. **b** the important habitat-forming corals *Acropora* spp. **c** and Pocilloporidae (**d**), and important algal groups: crustose coralline algae (CCA) (**e**), articulate calcareous Rhodophyta (Red) (**f**), non-calcareous Rhodophyta (**g**), and non-calcareous Phaeophyta (Brown) (**h**). Point colour represents station mean $\Omega_{Ar}$ (legend as per Fig. 2). The black line is the modelled mean, and the grey lines are 95% confidence intervals (CI). The red vertical line is the no-significant-effect concentration (NSEC). * denotes statistical significance in generalised linear models at $p < 0.05$ (Supplementary Table 2).

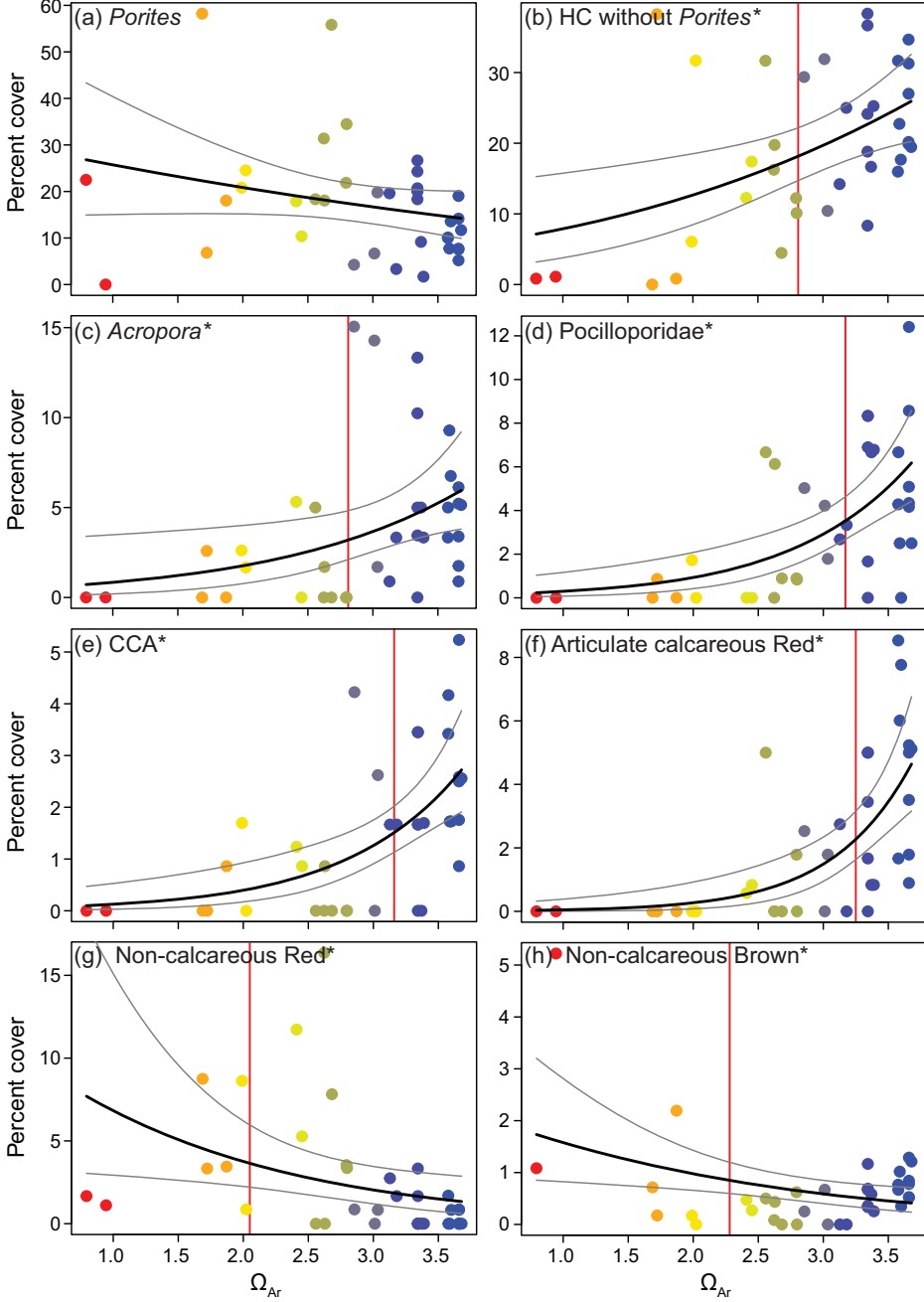

corals and CCA. The percent cover of all hard coral taxa abundant enough for analyses, except for massive *Porites* spp., was shown to be negatively affected by the increasing $\Omega_{Ar}$ gradient. Importantly, hard coral diversity, in both adult and juvenile life-stages, significantly declined with a $\Omega_{Ar}$ reduction of 0.3 units. At the present rate of change, these mean $\Omega_{Ar}$ values will be reached in the tropics by the year ~2060[6,7,39]. Reef communities continued to change below this $\Omega_{Ar}$ level, and a reduction of 0.5 $\Omega_{Ar}$ units from ambient values, as expected throughout the tropics by 2100 under the emissions scenario SSP2-4.5[6,39], was associated with considerable community change. By $\Omega_{Ar}$ 3.0, the percent cover of the highly susceptible calcareous red algae (both CCA and articulate calcareous Rhodophyta) halved, and the cover of structurally complex coral species (e.g. the Acroporidae and Pocilloporidae) reduced by 30%. These more sensitive corals were then almost absent from communities by $\Omega_{Ar}$ levels of 2.5, as predicted for the end of the century under emissions scenario SSP3-7.0. The mean $\Omega_{Ar}$ of contemporary reefs in Australia's Great Barrier Reef ranges by 0.5 units, and

reefs at the lower end of this gradient similarly record reduced CCA and coral juvenile abundances[16,36]. Hence the effects of reduced $\Omega_{Ar}$ are already manifest in some present-day reefs, and these patterns will strengthen as atmospheric $CO_2$ increases.

Previous studies examining changes to reef communities exposed to elevated $CO_2$ have also largely shown declines in calcareous taxa and concomitant increases in the non-calcareous[29,31,32,40]. The mechanisms responsible for community changes are not fully understood, as the response of communities to high $CO_2$ is often greater than the direct physiological responses seen for individual taxa. For example, metanalyses from Kornder et al.[19] report calcification rates are reduced by an average of 18% at $\Omega_{Ar} < 2.5$ for all corals combined. In contrast, in the present study, communities at $\Omega_{Ar}$ 2.5 exhibited a > 30% reduction in hard coral diversity and a decline of >50% cover of the more sensitive taxa compared to ambient values. Here high $CO_2$ communities were dominated by massive *Porites*, which can be important components of the reef framework[41], but lack the structural

## Table 1 | Estimates of changes in coral reef benthos along the aragonite saturation state ($\Omega_{Ar}$) gradient

| | $\Omega_{Ar}$ 3.5 | $\Omega_{Ar}$ 3.0 | $\Omega_{Ar}$ 2.5 | $\Omega_{Ar}$ NSEC |
|---|---|---|---|---|
| Quadrat complexity score (1–5)* | 3.07 (0.10) | 2.57 (0.08) | 2.15 (0.09) | 3.26 |
| Articulate calcareous Rhodophyta cover* | 3.45 (0.55) | 1.49 (0.34) | 0.63 (0.26) | 3.25 |
| Ratio calcareous : total algal biomass* | 0.54 (0.04) | 0.37 (0.04) | 0.23 (0.04) | 3.24 |
| HC adult diversity* | 10.55 (0.51) | 8.55 (0.36) | 6.93 (0.41) | 3.2 |
| HC juvenile diversity* | 5.24 (0.33) | 4.01 (0.22) | 3.07 (0.24) | 3.2 |
| HC juvenile density* | 10.74 (0.81) | 7.79 (0.54) | 5.65 (0.59) | 3.2 |
| Pocilloporidae cover* | 5.08 (0.72) | 2.91 (0.46) | 1.65 (0.44) | 3.17 |
| CCA cover* | 2.22 (0.33) | 1.26 (0.21) | 0.71 (0.20) | 3.16 |
| SC juvenile density* | 0.51 (0.11) | 0.24 (0.07) | 0.12 (0.06) | 3.12 |
| Complex HC cover* | 12.73 (1.48) | 8.57 (0.98) | 5.68 (1.04) | 3.11 |
| Seriatopora spp. cover* | 2.37 (0.48) | 1.19 (0.30) | 0.59 (0.27) | 3.11 |
| Fungiidae cover* | 0.88 (0.18) | 0.49 (0.11) | 0.27 (0.11) | 3.06 |
| SC juvenile diversity* | 0.42 (0.09) | 0.23 (0.06) | 0.13 (0.05) | 3.05 |
| Goniastrea spp. cover* | 2.22 (0.38) | 1.46 (0.25) | 0.96 (0.25) | 2.97 |
| Pocillopora spp. cover* | 2.67 (0.57) | 1.68 (0.36) | 1.05 (0.37) | 2.91 |
| Turbinaria spp. biomass* | 32.50 (4.36) | 23.49 (3.51) | 14.47 (3.95) | 2.88 |
| HC without Porites spp. cover* | 24.18 (2.58) | 19.69 (1.85) | 15.86 (2.06) | 2.81 |
| Acropora spp. cover* | 5.24 (1.00) | 3.66 (0.66) | 2.54 (0.70) | 2.81 |
| Acroporidae cover* | 10.28 (1.67) | 7.55 (1.13) | 5.51 (1.23) | 2.81 |
| Calcareous algae all biomass* | 65.52 (9.41) | 48.52 (7.56) | 31.53 (8.52) | 2.79 |
| Non-calcareous macroalgae cover* | 7.07 (1.09) | 9.14 (1.06) | 11.75 (1.21) | 2.74 |
| Non-calcareous algae biomass without Turbinaria spp. * | 16.07 (5.85) | 25.81 (4.70) | 35.55 (5.30) | 2.73 |
| Calcareous algae cover* | 11.70 (1.60) | 9.40 (1.12) | 7.51 (1.24) | 2.68 |
| Total macroalgae biomass* | 114.08 (11.32) | 97.82 (9.09) | 81.55 (10.24) | 2.6 |
| Merulinidae cover* | 6.01 (1.08) | 4.69 (0.74) | 3.63 (0.81) | 2.59 |
| Melanamansia spp. biomass* | 5.70 (2.29) | 8.76 (1.85) | 11.83 (2.08) | 2.53 |
| Non-calcareous Phaeophyta cover* | 4.53 (1.12) | 5.88 (1.10) | 7.60 (1.27) | 2.28 |
| Sponge cover* | 1.68 (0.40) | 2.13 (0.39) | 2.70 (0.45) | 2.19 |
| Non-calcareous Rhodophyta cover* | 1.50 (0.52) | 2.04 (0.54) | 2.78 (0.63) | 2.05 |
| Hard coral cover | 38.60 (2.63) | 37.13 (2.05) | 35.68 (2.29) | - |
| Massive Porites spp. cover. | 14.86 (2.34) | 16.68 (1.99) | 18.68 (2.23) | - |

Parameters are ranked in order of susceptibility to $\Omega_{Ar}$ changes based on no-significant-effect-concentrations (NSEC). Mean values (±SE) of the benthic cover (%) of hard (HC) and soft (SC) corals and other benthos, the density (colonies m$^{-2}$) and diversity (genera m$^{-2}$) of hard and soft adult and juvenile corals, and the biomass (g m$^{-2}$) of the algal communities are shown at $\Omega_{Ar}$ 3.5, 3.0 and 2.5. All data are presented on the measured scale. *indicates a significant response to $\Omega_{Ar}$ in the generalised linear model at $p < 0.05$ (Supplementary Table 2).

complexity to provide shelter for many habitat-associated species[42–44]. The observed declines in CCA and proliferation of Melanamansia spp. and other non-calcareous macroalgae at low $\Omega_{Ar}$ also likely impacted coral communities, as CCA facilitates coral recruitment while macroalgae hinders it[45,46]. The differential responses of certain algae, for example the decline in the non-calcareous Turbinaria spp. and relative robustness of the lightly calcareous Peyssonnelia spp. at high $CO_2$, require further investigation. The seep communities are likely shaped not only by the direct physiological effects of elevated $CO_2$ on certain taxa, but also by indirect ecological effects that can have substantial impact and are difficult to predict based on physiological studies alone[42,43,47].

Our empirical data show that the magnitude of change to coral reef communities attributable to ocean acidification by 2100 will strongly depend on $CO_2$ emissions. By 2050, $\Omega_{Ar}$ will likely be 0.2 units lower than at present[7], but by 2100, levels will vary greatly depending on the SSP scenario realised[6,39]. Under the most optimistic IPCC scenario (SSP1–1.9), where $CO_2$ emissions are cut to net zero by 2050 and atmospheric $CO_2$ will have slightly eased, coral reefs in 2100 will not be altered greatly by ocean acidification in comparison to the present day (Fig. 6). Alternatively, the most pessimistic scenario (SSP5–8.5), where present-day annual $CO_2$ emissions

triple by 2100, would result in drastic changes to reef communities. It is becoming increasingly unlikely that either of these extreme scenarios will occur, with more recent estimates suggesting that SSP3-7.0 may be most appropriate in impact assessment studies[48]. Under this scenario, 2100 coral reefs would see considerable declines in hard coral diversity (40%), the percent cover of CCA (70%) and complex coral species (60%), and drastic increases in the cover of non-calcareous macroalgae (80%) (Fig. 6). If emissions are limited to the more conservative SSP2–4.5, we may still see a 50% loss in CCA cover and a 40% increase in the cover of non-calcareous macroalgae by 2100. Importantly, these predictions consider the effects of ocean acidification in isolation, and coral reefs are under increasing pressure from a range of sources. Global mass coral bleaching events from marine heat waves, also driven by anthropogenic $CO_2$ emissions, and other forms of disturbances from climate change, are increasing in frequency and intensity and are increasingly impacting the world's coral reefs[49].

These $CO_2$ seep sites are not perfect representations of future coral reefs. They are small in size and lack the co-occurring elevated temperature stress expected under climate change. Seep $CO_2$ levels are also characteristically variable, especially within areas with low mean pH[29] (Supplementary Fig. 1), and effects of this variability are largely unknown for most coral

**Fig. 4 | Changes in the diversity and abundance of hard and soft corals in relation to aragonite saturation state ($\Omega_{Ar}$).** The diversity of adult hard corals (HC) (**a**), juvenile hard corals (**b**), adult soft corals (SC) (**c**), juvenile soft corals (**d**), and the density of juvenile hard (**e**) and soft corals (**f**) in relation to station aragonite saturation state ($\Omega_{Ar}$). See Fig. 3 legend for plot description.

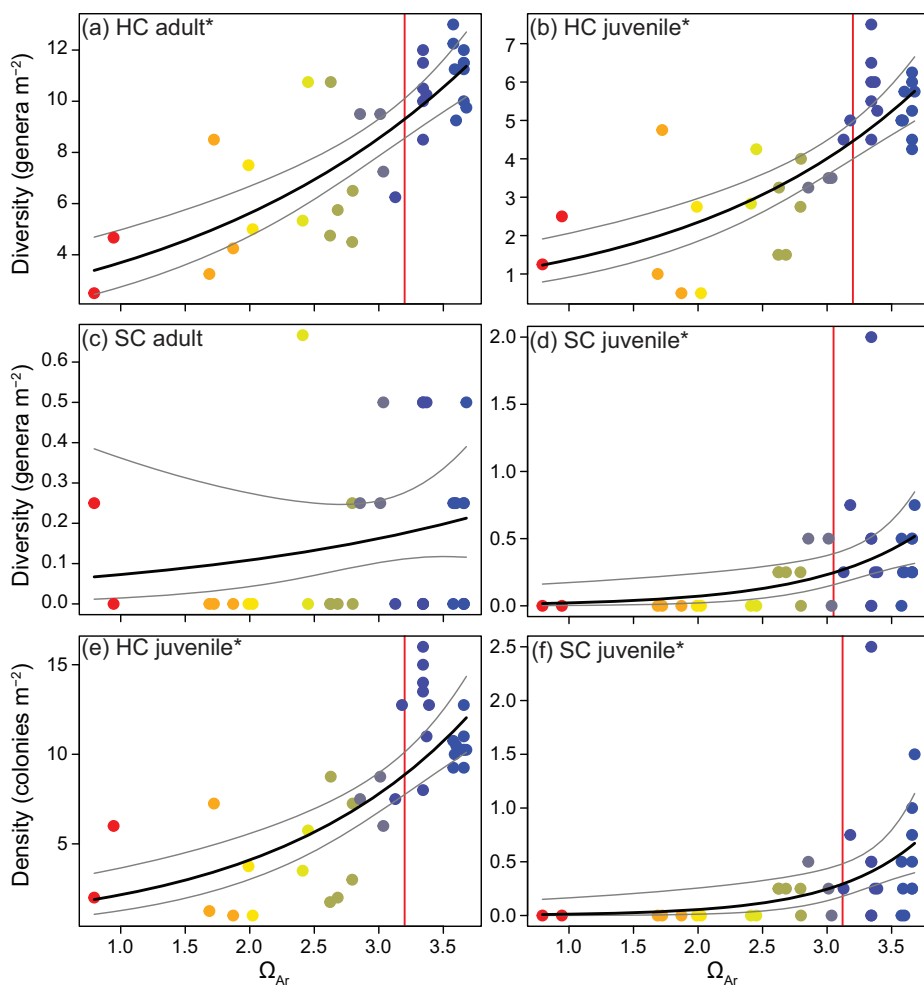

reef organisms[50]. Hence extrapolating the results seen here to the future of coral reefs globally will attract some uncertainty. However, studies in situ at $CO_2$ seeps have advantages over laboratory studies (e.g. organisms grow under natural conditions of ecological interactions and have life-time acclimation) and hence provide unique insights into the effects of ongoing ocean acidification on marine communities.

Anthropogenic ocean acidification has been ongoing since the start of the Industrial Revolution, and global $CO_2$ emissions continue to rise largely unabated[6]. Reliable in situ data are essential to validate observed changes from ocean acidification that have likely already occurred (but are not usually factored into models or debates), and to predict how they will continue to impact coral reefs[6,16,51]. Our data clearly show that these changes will intensify in the coming decades, with reef communities shifting away from dominance by many scleractinian corals towards less diverse and structurally complex communities characterised by fewer tolerant species and non-calcareous taxa, with a raft of flow-on effects on reef-associated organisms[42,43]. The magnitude of this change will depend on our $CO_2$ emissions pathway, with more emissions resulting in a greater change to present-day coral reefs. Reef resilience and recovery following disturbances will also decline, as ocean acidification reduces rates of coral recruitment[52]. Reductions in atmospheric $CO_2$ levels are urgently needed to prevent further deviations from the reefs of today.

## Methods
This study was conducted at the volcanic $CO_2$ seep at Upa-Upasina, Normanby Island, Milne Bay, Papua New Guinea (PNG). The $CO_2$ seep site has been described in detail by Fabricius et al.[29]. It is characterised by a mosaic of nearly pure $CO_2$ gas seeping through the sea floor in patches of varying intensity in the shallow waters along several hundred metres of coastline. This $CO_2$ locally alters the carbon chemistry of the seawater without altering the temperature or salinity[24,44]. Seeping intensity is characteristically variable over short timeframes (minutes to hours), but mean $CO_2$ levels around the seep site have remained relatively constant over 6 years of sampling (e.g. Supplementary Table 2 and Supplementary Fig. 3 from[29] and the present study). To reach areas unaffected by the volcanic $CO_2$, stations were spread well beyond the area of visible seeping, up to ~500 m to the south and north of the seep along the same island fringing reef. Sampling stations were established at ~3 m depth, each marked with a small sub-surface float ($n = 37$). Stations were spread widely across and along the seep seascape, to capture varying seep intensities.

The carbon chemistry of each of the 37 stations was characterised in two ways. Firstly, bottle samples were collected ~twice-daily for 2 weeks at each station for pH and total alkalinity ($A_T$). Where $CO_2$ was elevated, one sample per station was taken during each sample period (pH $n = 17$–18, $A_T$ $n = 3$–4 per station). Of the fifteen stations unaffected by volcanic $CO_2$, six were investigated for their carbon chemistry (pH $n = 11 – 17$, $A_T$ $= 3$–4), and the remaining nine stations were given the average carbon chemistry values of the adjacent ambient stations in further analyses. All pH samples were measured within 1 h of collection using a benchtop pH electrode (Eutech) and meter (Oakton, USA), with mV converted to $pH_{Total}$ (hereafter $pH_T$) by comparison to readings from a certified TRIS pH buffer (A.G. Dickson lab, Scripps Institution of Oceanography, USA) following standard procedures[53]. Samples for $A_T$ were preserved with saturated $HgCl_2$ and processed at the laboratories of the Australian Institute of Marine Science

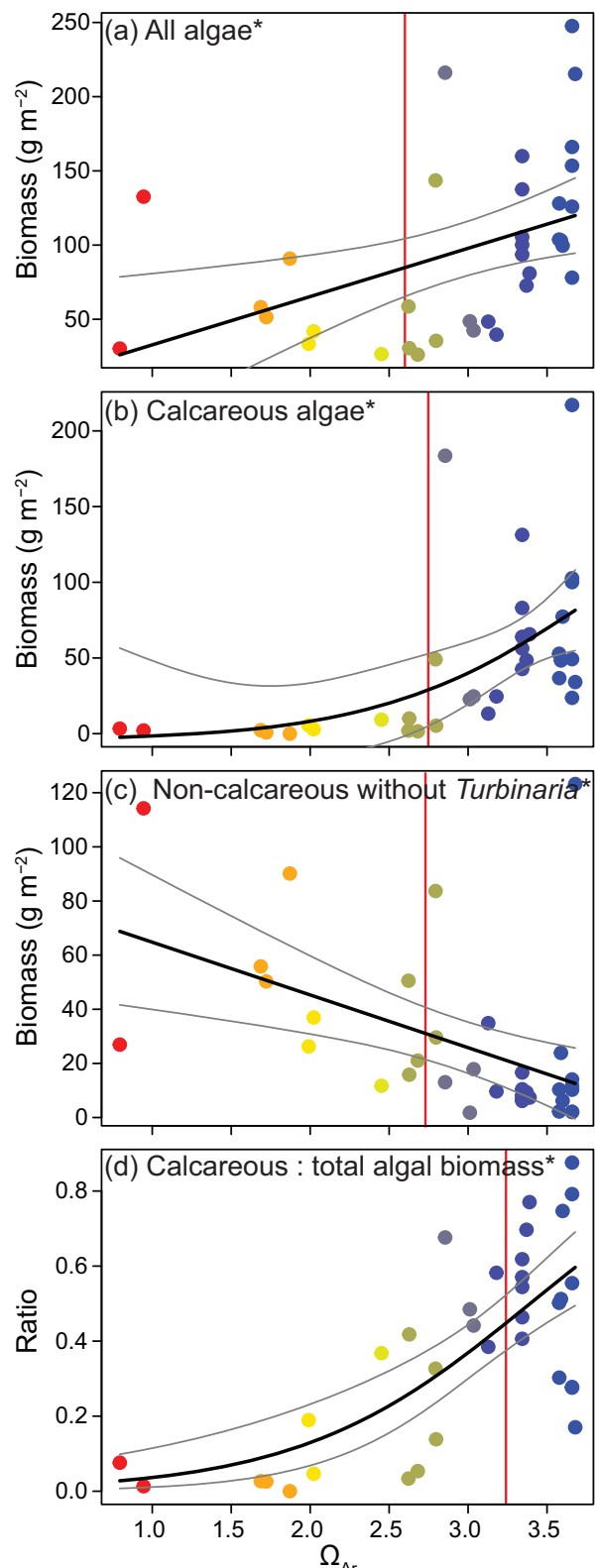

**Fig. 5 | Changes in macroalgal biomass in relation to aragonite saturation state ($\Omega_{Ar}$).** The biomass of all macroalgae (**a**), calcareous macroalgae (**b**), total non-calcareous macroalgae without *Turbinaria* spp. **c** and the ratio of calcareous (Calc) to total macroalgal biomass (**d**) in relation to station aragonite saturation state ($\Omega_{Ar}$). See Fig.3 legend for plot description.

(AIMS) within 1 month of collection using open cell potentiometric titration (855 robotic titrosampler, Metrohm) following Dickson et al.[53]. $A_T$ values were calculated using a Gran function and comparisons to certified reference materials were <5 μmol kg⁻¹ (CRM batch 141, A.G. Dickson lab, Scripps Institution of Oceanography, USA). Secondly, pH loggers (SeaFET, Seabird USA) were deployed for ~24 h at each station, sampling every 10 min. The two pH loggers rotated through all but three of the $CO_2$ affected stations (n = 19), and two of the stations unaffected by volcanic $CO_2$. The pH loggers were also deployed side-by-side away from the seep in ambient seawater three times throughout the sampling period, and an offset was applied to one instrument to account for differences between loggers (~0.07 pH$_T$) after comparisons with parallel bottle pH samples (n = 3). Estimates of station pH derived from the bottle measurements and SeaFET loggers were highly correlated (Supplementary Fig. 1) and subsequently combined for further analyses. To do so, pH data were averaged for each method after being standardised by the square root of their sample size. Here sample size was either the number of bottle samples, or the number of 10 min readings the SeaFET took over the ~24 h deployment at each station. This weighted approach avoided either dataset obscuring the other. Measured pH$_T$ and $A_T$ values were used to calculate the remaining carbon chemistry variables for each station using the CO2SYS macro[54] with the constants of Dickson and Milero[55] (Supplementary Table 1).

At each station, a series of 1 m² quadrats were sampled on SCUBA to characterise benthic communities. Four quadrats were sampled at each station, however five stations unaffected by volcanic $CO_2$ had only two quadrats sampled. Quadrats were placed immediately around the sub-surface floats, to ensure the measured carbon chemistry parameters from the floats were representative of the quadrat area. Quadrat placement avoided areas completely occupied by single large coral colonies, likely underrepresenting large table *Acropora* corals commonly found at the sites unaffected by volcanic $CO_2$, as well as massive *Porites* spp. which dominate the seep[29]. Photographs were taken of each quadrat for analysis of benthic cover and community composition[56]. To do so photographs were overlaid with 28 evenly spaced points and the benthos underneath each was recorded to the highest taxonomic level possible (a total of 4004 observations). This was genus level for hard corals and soft corals, while macroalgae and other taxa were classified into phyla. Organisms were also categorised as being calcareous or not; calcareous being those with solid calcium carbonate supporting structures (e.g. the scleractinian corals and CCA), while non-calcareous lack these heavy deposits but includes some organisms that contain small amounts of $CaCO_3$ (e.g. *Peyssonnelia* and *Padina* spp. algae and *Sarcophyton* soft corals). The morphology of hard corals was also recorded (e.g. branching *Acropora* spp., tabular *Acropora* spp. etc.). Visual in situ surveys were conducted in each quadrat for juvenile (<5 cm diameter) hard and soft coral density and diversity[56]. Each quadrat was further visually classified for structural complexity on a scale of 1–5, with 1 being least complex and 5 being highly complex[43]. Each type of survey was conducted by a single observer to minimise bias.

All macroalgae occurring within one quarter (i.e. 0.25 m²) of two of the quadrats per station were hand-collected by divers using a scraping tool. Larger areas were not sampled due to time constraints in the field. Much of the macroalgae were cryptic, occurring within the crevices of the reef where they were protected from grazing, and hence unlikely to be documented in the photo-surveys. Collections did not sample encrusting algal taxa, such as crustose coralline algae (CCA) and *Peyssonnelia* spp., nor turfs. Macroalgae samples were sundried, transported to AIMS, sorted into calcareous and non-calcareous taxa, and after 3 days in a drying oven at 60°C, weighed to the nearest 0.1 mg (Shimadzu AW220, Japan). The two most dominant genera (*Melanamansia* and *Turbinaria*) were also separated and weighed.

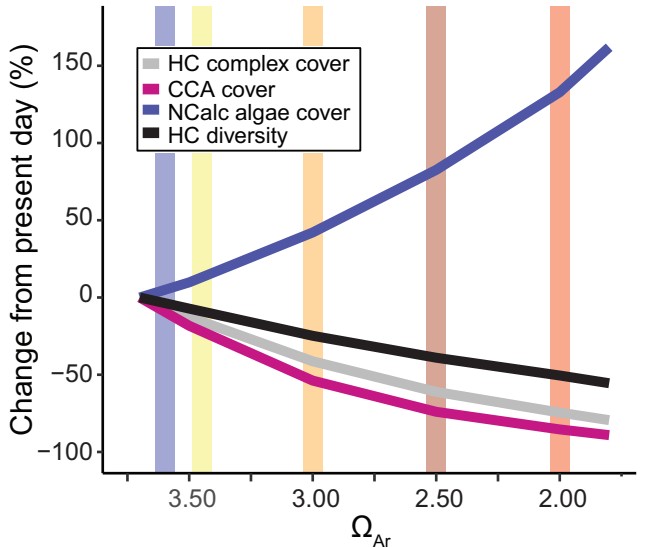

**Fig. 6 | Observed changes in key reef benthos in relation to the saturation state of aragonite ($\Omega_{Ar}$).** Shown are the percent cover of complex hard corals (HC, light grey), crustose coralline algae (CCA, magenta), non-calcareous (NCalc) macroalgae (blue), and hard coral diversity (black). The vertical lines represent the $\Omega_{Ar}$ values predicted for much of the tropics for the year 2100 by Jiang et al.[39] for the five Shared Socioeconomic Pathways emission scenarios: SSP1–1.9 (blue), SSP1–2.6 (yellow), SSP2–4.5 (orange), SSP3–7.0 (brown) and SSP5–8.5 (red)[6,39].

## Statistics and reproducibility

All statistics were conducted in R (version 4.4.0)[57]. $\Omega_{Ar}$ values calculated from $pH_T$ and $A_T$ were chosen as predictors as they can strongly influence the physiology of corals and other calcareous reef taxa and were highly correlated with other carbon chemistry variables (Fig. 1). All benthic data (i.e. benthos percent cover, coral juvenile abundance and diversity, algal biomass etc.) were averaged across quadrats by station, prior to statistical analyses, as each station was represented by a single $\Omega_{Ar}$ value in models. Changes in the cover of the communities were first examined via redundancy analysis (RDA). This allowed the visualisation of multi-dimensional community data in two-dimensional space, and to test the significance of mean station $\Omega_{Ar}$ (continuous variable) on community changes via PERMANOVA.

Generalised linear models (GLM) were used to examine the changes in the percent cover of different taxa from the photo surveys, the density and diversity of hard and soft coral adult and juvenile corals, and biomass of the algal communities in relation to mean $\Omega_{Ar}$ from each of the stations. Different link functions were used in models depending on data type (Supplementary Table 2): quasibinomial for percent cover, quasipoisson for juvenile abundance and diversity counts, and gaussian for algal biomass weights. Algal weights were square-root transformed to better fit the assumed distribution, with the final model chosen based on lowest Akaike information criterion (AIC). The "Predict" function in R was used to estimate response values at $\Omega_{Ar}$ 3.5, 3.0 and 2.5, approximately representing $\Omega_{Ar}$ ambient values away from the seeps, and what is projected for tropical coral reefs in the region by the year 2100 under the Intergovernmental Panel on Climate Change's (IPCC) shared socio-economic pathways SSP2-4.5 and SSP3-7.0[6,39]. No-significant-effect concentration (NSEC) values were also calculated for each GLM following Fisher and Fox[15]. Here the NSEC is the $\Omega_{Ar}$ where the predicted response first significantly deviates above or below (depending on the shape of the response curve) the response at an ambient $\Omega_{Ar}$ of 3.5.

## Reporting summary

Further information on research design is available in the Nature Portfolio Reporting Summary linked to this article.

## Data availability

Numerical source data underlying the main and Supplementary Figs. are included in Supplementary Data files 1–4 and are available via the Australian Institute of Marine Science data repository: https://apps.aims.gov.au/metadata/view/76c40b3c-8535-43b6-bf5f-ad26d6e1ad92.

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

## Acknowledgements

We thank Dr Riccardo Rodolfo-Metalpa, L'Institut de Recherche pour le Développement, Noumea, New Caledonia, for co-funding the fieldwork through a grant by L'Agence nationale de la recherche, for collaboration and logistic support. We thank the community of Illi Illi Bwa Bwa (Upa Upasina) for allowing us access to their unique reefs, and the crew of the MV Chertan for supporting the field work. This project was funded by the Australian Institute of Marine Science.

## Author contributions

SN and KF conceived the study and conducted the fieldwork. SN and CB completed the lab work and image analyses. SN and RF conducted statistical analyses. SN wrote the first draft of the manuscript. All authors provided input to the writing of later manuscript drafts.

## Competing interests

The authors declare no competing interests.
