## [Transparent Peer Review file · Communications Biology]

Progressive changes in coral reef communities with increasing ocean acidification

Corresponding Author: Dr Sam Noonan

This manuscript has been previously submitted at another journal. This document only contains information relating to versions considered at Communications Biology.

Version 0:

Reviewer comments:

Reviewer #1

(Remarks to the Author)

The manuscript by Noonan et al describes the changes in coral reef communities at one site across a gradient of changes in carbonate chemistry. These sites all have their inherent issues that it is not worth raising again, but also have benefits for the scientific community to learn from as unique systems. Importantly, the authors here demonstrate changes in coral reef communities across a gradient in seawater carbonate chemistry where the mean declines match those that will occur in the future due to ocean acidification. For the first time, they describe changes in community cover that can be tied to specific changes in mean pH/saturation state (across meaningful scales), rather than displaying results of control v impacted sites. Overall, the manuscript will aid the scientific community in understanding when the impacts of ocean acidification will occur and to what magnitude (given site specific caveats). I have no major issues with the ms, beyond those already raised numerous times that don't need repeating, and the fact that perhaps the extrapolation at the end is over reaching. The issues are outweighed by the utility of the information provided. Congrats to the authors here.

Major comments:

Why the focus on saturation state throughout? There some statement in the ms about this, but I guess also its probably not important, as changes in saturation state here largely correlate with changes in pH and not AT.

Specific comments:

Line 9: articulate "coralline algae"?

Line 10: Just state the specific percent values here if possible.

Line 70: I agree with the entire introduction. However, I wonder at this point whether the main difference between this manuscript and others could be emphasised for the naïve reader? E.g., this study provides the only decent estimate of what specific pCO₂ level causes change and to what extent? I know the authors mention this, but I suspect the lack of emphasis on this over the wide introductory material could lead a naïve reader into missing this point.

Line 83: What is the cause of increasing TA?

Fig 2: but peyssonieellas are calcifying reds? certainly differential responses at this site to the other calcareous reds, both here and in this group's prior work here. Some variability in community structure not caused solely by carbonate chemistry, as RDA 2 has no strong correlation.

Line 121: "spp." at the start of the sentence is accidentally in italics.

Line 222: Yes, but the data here are much more robust than those data presented in the references supplied on this line. The key issue is that prior tropical research simply has not properly quantified changes in communities with gradients of carbonate chemistry. The older references are perhaps outdated, given we know many coral species can indeed grow under low pH/saturation state.

Line 235: Yes, and most ecological interactions measured (if not all) have been post 1850 and most inference made with no thought about carbonate chemistry until around the late 90s.

Line 264: The issue with meta-analyses from lab studies is that they don't include ecological interactions or recruitment dynamics.

Line 273: I find it difficult to extrapolate between the results at one field site to all coral reefs. I agree that generally these changes are represented at other vent sites, but there are discrepancies between studies/sites with respects to what corals

remain, whether soft corals increase or decrease, and the role of macroalgae and sponges in future tropical oceans. Discussion: I was surprised that there was not a great deal of discussion into why *Turbinaria* declined so drastically and the red alga *Melanamansia* increased in abundance? It would be useful to describe their physiology better in future work if possible.

Line 339: is it possible to display these data anywhere? One of the largest critiques of CO₂ seep/vent work is the correlation between decline in mean pH coinciding with increases in variability in pH. If these data are not convincing, then authors should include some caveats in the discussion regarding the possible effects of variability in carbonate chemistry.

Reviewer #2

(Remarks to the Author)

Summary of Manuscript:

This manuscript reports the results of a mensurative survey of 37 sites in Papua New Guinea that are characterized by the presence and intensity of volcanic-induced carbon dioxide seeps into the shallow reef environment. Seeps alter the seawater chemistry in similar ways to how ongoing ocean acidification from the influx of atmospheric CO₂ affects oceans worldwide. These changes in SW chemistry can be represented by changes in the saturation state of aragonite [Ω_{Ar}] (the most common crystalline form of calcium carbonate in the skeletons of many tropical calcifiers, especially reef corals). The authors take advantage of this seep system to establish sites representing a gradient in Ω_{Ar} and quantify the seawater chemistry at each site, ecological aspects of the coral reef communities at each site (including benthic cover by coral and other calcifying taxa, benthic macroalgae, and other benthic taxa), diversity of corals, biomass of macroalgae, and several other community characteristics. Over a range of Ω_{Ar} from 0.79 to 3.57, the effects of reduced Ω_{Ar} had predictable effects on seawater chemistry, calcifier abundance, diversity, and macroalgal biomass and these effects largely were consistent with previous results of both in situ studies and mesocosm experiments. As there were no obvious tipping points (non-linear responses), the authors used their results to predict longer-term responses of reef communities to ongoing ocean acidification [OA] under different CO₂ emission scenarios.

Overall Impression:

The authors should be commended on the scope and scale of their study. While the overall approach is not novel, what is novel is the use of an extensive natural gradient in Ω_{Ar} to address a wide range of seawater conditions and a plethora of reef community responses. Given that this study was conducted in a confined region with the same species pool their results are not confounded by biogeographic effects. If valid, these results can be used in a predictive framework to address anticipated future changes in reef communities to OA. As with any study, there are some limitations on the interpretation of results and perhaps some methodological issues to address. The data figures and tables are clear and useful for interpreting the results of the study and the supplementary material provides adequate details to evaluate their interpretation. Overall, I believe that subject to the comments below, the results of this study will be widely appreciated by coral reef scientists.

Specific Comments:

1) It is useful that this gradient in SW chemistry can be addressed over such a relatively small spatial scale. However, given that the study was conducted at one point in time (days/weeks/months), it does raise the question of how constant these seeps (and SW chemistry at each site) are over time. Do individual seeps vary in intensity (volume emitted per time) over time? Surely at some time scale they much vary. But is this over days/weeks/years/millennia? Is there interaction among seep sites as the result of water flow and mixing? Again, this would alter seawater chemistry (and its effects) over different scales. Any variation in seep intensity would have differential effects on SW chemistry (short-term) and community attributes (longer-term). So any data (or references) addressing how stable individual seep sites are, would be needed to evaluate the community-level effects inferred in this study.

2) The benthic sampling that was conducted at each site needs some justification. Benthic composition (percent cover, diversity, etc.) at each site was quantified primarily from photographs of 4, 1-m² quadrats at each site. This strikes me as a fairly low sample size for coral reefs which are notably for their patchiness. Furthermore, these ecological data were extracted from the photographs by overlaying 28 randomly-located dots to determine taxa abundances. This too is a modest number of sampling points per photograph for such a diverse community. Often 100s of points are used per photograph and many coral reef studies.

3) The sampling of macroalgae (abundance and biomass) was conducted in 2, 0.25-m² sub-quadrats within only 2 of the 4 larger quadrats at each site. This too is a pretty small sample size for a diverse community. I can appreciate the need to allocate field efforts to various tasks, but undersampling can have insidious effects on results. So I think some justification would be useful.

4) It probably is not a big deal, but the macroalgae were initially sun-dried in the field and then dried in an oven later at 45 degrees C. Why so low? Typically algae are dried at 60-80 degrees C to quantify dry mass.

5) I don't intend for the above comments to result in "death by a thousand cuts", but I do think that these issues need to be addressed in order for the reader to truly appreciate the importance of the effects that are presented.

Reviewer #3

(Remarks to the Author)

Review of Noonan et al Comms Bio

This paper looks at coral reef benthic community structure (hard and soft coral, calcareous and non-calcareous) at 37 stations along a gradient of aragonite saturation state at a CO₂ seep in PNG. Saturation state varied from 3.7 at the control sites to 0.8 at the most impacted sites and pH ranged from 8.0 to 7.3. The manuscript is well written and does a good job over covering the current literature on the topic. The authors report that percent cover of structurally complex (branching, corymbose and tabulate) coral and calcareous algae decreased by 50% across the gradient. Some taxa, showed significant reductions when the saturation state had just dropped from 3.7 to 3.2. Non-calcareous red and brown increased in cover over the same gradient. Notably these changes did not exhibit any clear tipping points. This study represents the most rigorous analysis to date of a coral reef site impacted by a CO₂ seep and it provides valuable insights into how non-CO₂ seep coral reefs may be impacted by ocean acidification in coming decades. The authors point out that based on these results significant impacts of OA can be expected by the year 2100 for the two middle of the road CO₂ emission scenarios (SSP2-4.5 and SSP3-7.0). The authors rightly point out that these projections are based on the effect of saturation state reduction alone and ignore the likely significant interactions resulting from temperature increases that will happen in parallel with decreasing pH. The statistical analysis looks appropriate and correct to me. The collection and analysis of the chemical data is appropriate. Although I would recommend that the authors switch to spectrophometric pH for future studies to take advantage of much greater precision and accuracy. This is a valuable contribution to the literature on ocean acidification effects on coral reef community structure.

Version 1:

Reviewer comments:

Reviewer #1

(Remarks to the Author)

The authors have done an excellent job revising the manuscript. Great work, and congrats on an excellent paper. Chris C.

Reviewer #2

(Remarks to the Author)

In my opinion, the authors have done a very good job of addressing my earlier questions and concerns. I am now supportive of this manuscript moving forward for publication.

Open Access This Peer Review File is licensed under a Creative Commons Attribution 4.0 International License, which permits use, sharing, adaptation, distribution and reproduction in any medium or format, as long as you give appropriate credit to the original author(s) and the source, provide a link to the Creative Commons license, and indicate if changes were

made.

Reviewer 1

Comment	Response
Why the focus on saturation state throughout? There some statement in the ms about this, but I guess also its probably not important, as changes in saturation state here largely correlate with changes in pH and not AT.	Aragonite saturation state was used as a predictor in models over other carbon chemistry variables as aragonite is the CaCO_3 mineral form found in most coral reef calcifiers (eg the scleractinian corals) and aragonite saturation variation has been shown to affect the physiology of many reef organisms. Using aragonite saturation allowed us to directly compare our results with other similar studies that have examined reef community changes with increasing ocean acidification (eg Smith et al. 2020, Kleypas et al. 1999). This justification is given in the text (see below excerpt). Furthermore, data for all carbon chemistry parameters from each station are reported in the supplement (Table S1) and the community composition and carbon chemistry data will become available for download via the link in the data accessibility statement upon publication. Line 397: “Ω_{Ar} values calculated from pH_T and A_T were chosen as predictors as they can strongly influence the physiology of corals and other calcareous reef taxa and were highly correlated with other carbon chemistry variables”
Line 9: articulate “coralline algae”?	The suggested edit has been made.
Line 10: Just state the specific percent values here if possible.	Unfortunately the word count limit for the abstract does not allow us to list each individual percentage change in this instance. The suggested change has not been made.
Line 70: I agree with the entire introduction. However, I wonder at this point whether the main difference between this manuscript and others could be emphasised for the naïve reader? E.g., this study provides the only decent estimate of what specific pCO_2 level causes change and to what extent? I know the authors mention this, but is suspect the lack of emphasis on this over the wide introductory material	We thank the reviewer for the comment and have adjusted the section to emphasise the novelty of our data. We have added to Line 70: “For the first time, these response curves allowed us to quantify the minimum CO_2 increases that resulted in significant reef functional changes, and to project future changes to key coral reef benthos under different CO_2 emission scenarios.”

could lead a naïve reader into missing this point.	
Line 83: What is the cause of increasing TA?	We believe the increasing total alkalinity is due to CaCO₃ dissolution. This information is given on line 86. “...but increased relative to pH_T at pH_T values below ~7.7 due to their elevated A_T, likely due to increased CaCO₃ dissolution.”
Fig 2: but peyssonniellas are calcifying reds? certainly differential responses at this site to the other calcareous reds, both here and in this group’s prior work here. Some variability in community structure not caused solely by carbonate chemistry, as RDA 2 has no strong correlation.	We agree that Peyssonnelia spp. contain a small amount of calcium carbonate. However this is far less than that of the organisms we defined as calcareous in this study (e.g. scleractinian corals and CCA). Other organisms that are similarly soft bodied (e.g. Padina spp. algae and Sarcophyton spp. soft corals) were also defined as non-calcareous. This definition has now need clarified in the Methods section on Line 377: “Organisms were also categorised as being calcareous or not; calcareous being those with solid calcium carbonate supporting structures (e.g. the scleractinian corals and CCA), while non-calcareous lack these heavy deposits but includes some organisms that contain small amounts of CaCO₃ (e.g. Peyssonnelia and Padina spp. algae and Sarcophyton soft corals).” We have also now acknowledged in the Discussion that Peyssonnelia spp. are robust to CO₂ and their distribution patterns require further investigation. This sentence has been added to Line 272: “The differential responses of certain algae, for example the declines in the non-calcareous Turbinaria spp. and relative robustness of the lightly calcareous Peyssonnelia spp. at high CO₂, requires further investigation.”
Line 121: “spp.” at the start of the sentence is accidentally in italics.	This typo has been corrected.
Line 222: Yes, but the data here are much more robust than those data presented in the references supplied on this line. The key issue is that prior tropical research simply has not properly quantified changes in communities with gradients of carbonate chemistry. The	We agree there have been few investigations to date with enough treatment levels to investigate the effects of gradients of CO₂ exposure on coral reefs. Those cited here are the most relevant comparisons to the present study we are aware of. The older references have been removed from the comparison.

older references are perhaps outdated, given we know many coral species can indeed grow under low pH/saturation state.	
Line 235: Yes, and most ecological interactions measured (if not all) have been post 1850 and most inference made with no thought about carbonate chemistry until around the late 90s.	Agreed.
Line 264: The issue with meta-analyses from lab studies is that they don't include ecological interactions or recruitment dynamics.	Agreed. This is acknowledged in the Discussion. Line 274: "The seep communities are likely shaped not only by the direct physiological effects of elevated CO₂ on certain taxa, but also by indirect ecological effects that can have substantial impact and are difficult to predict based on physiological studies alone"
Line 273: I find it difficult to extrapolate between the results at one field site to all coral reefs. I agree that generally these changes are represented at other vent sites, but there are discrepancies between studies/sites with respects to what corals remain, whether soft corals increase or decrease, and the role of macroalgae and sponges in future tropical oceans.	We agree that scaling up the results of this study to the future of coral reefs globally will attract some uncertainty. This has been acknowledged in the Discussion. The following paragraph has been added from Line 305: "These CO₂ seep sites are not perfect representations of future coral reefs. They are small in size and lack the co-occurring elevated temperature stress expected under climate change. Seep CO₂ levels are also characteristically variable, especially within areas with low mean pH¹ (Fig. S1), and effects of this variability are largely unknown for most coral reef organisms². Hence extrapolating the results seen here to the future of coral reefs globally will attract some uncertainty. However, studies in situ at CO₂ seeps have advantages over laboratory studies (e.g. organisms grow under natural conditions of ecological interactions and have life-time acclimation) and hence provide unique insights into the effects of ongoing ocean acidification on marine communities."
Discussion: I was surprised that there was not a great	We agree these patterns are of great interest, but the mechanisms responsible remain largely

deal of discussion into why Turbinaria declined so drastically and the red alga Melanamansia increased in abundance? It would be useful to describe their physiology better in future work if possible.	unknown. We have now better highlighted this in the discussion. Line 272: “The differential response of certain algae, for example the declines in the non-calcareous Turbinaria spp. and relative robustness of the lightly calcareous Peyssonnelia spp. at high CO₂, requires further investigation.”
Line 339: is it possible to display these data anywhere? One of the largest critiques of CO₂ seep/vent work is the correlation between decline in mean pH coinciding with increases in variability in pH. If these data are not convincing, then authors should include some caveats in the discussion regarding the possible effects of variability in carbonate chemistry.	We agree that the seep site carbon chemistry is more variable than many coral reefs, and that the effects of this is largely unknown. We have shown the variation in station carbon chemistry through the error bars in Fig. S1. As expected, this variability becomes greater as the pH decreases. A paragraph has been added to the discussion to acknowledge this variability and its uncertain effects. Line 305: “These CO₂ seep sites are not perfect representations of future coral reefs. They are small in size and lack the co-occurring elevated temperature stress expected under climate change. Seep CO₂ levels are also characteristically variable, especially within areas with low mean pH¹ (Fig. S1), and effects of this variability are largely unknown for most coral reef organisms². Hence extrapolating the results seen here to the future of coral reefs globally will attract some uncertainty. However, studies in situ at CO₂ seeps have advantages over laboratory studies (e.g. organisms grow under natural conditions of ecological interactions and have life-time acclimation) and hence provide unique insights into the effects of ongoing ocean acidification on marine communities.”

Reviewer 2

1)It is useful that this gradient in SW chemistry can be addressed over	It is well known that the carbon chemistry of all seep sites globally is characteristically variable over short time frames (minutes to hours). We have
--	--

such a relatively small spatial scale. However, given that the study was conducted at one point in time (days/weeks/months), it does raise the question of how constant these seeps (and SW chemistry at each site) are over time. Do individual seeps vary in intensity (volume emitted per time) over time? Surely at some time scale they much vary. But is this over days/weeks/years/millennia? Is there interaction among seep sites as the result of water flow and mixing? Again, this would alter seawater chemistry (and its effects) over different scales. Any variation in seep intensity would have differential effects on SW chemistry (short-term) and community attributes (longer-term). So any data (or references) addressing how stable individual seep sites are, would be needed to evaluate the community-level effects inferred in this study.

some limited and anecdotal evidence to support our argument that our measurements are indicative of longer-term exposure:

We used stable isotope measurements in the skeletons of massive *Porites* corals to investigate their long-term exposure to CO₂ (Stewart Fallon, unpublished data, and methods section of ³). These data show remarkable stability in seep seawater CO₂ incorporated into the skeletons of *Porites* since at least the 1910's (Figure 1) concurrent with our seawater measurements. Here mean seawater pH has ranged by <0.15 units over the ~100-year time series.

Figure 1: Derived seawater pH from ¹⁴C radiocarbon measurements from the skeleton of a massive *Porites* coral skeleton core (ILI 35) taken within the Upa-Upasina seep site of the present study. The shaded horizontal bar represents adjacent control site pH. Error bars are standard errors.

We also have seen measured mean CO₂ values remaining relatively stable within this study site over the six years we have worked there. We have repeatedly used both discrete samples and loggers sampling at high frequency to determine the carbon chemistry of the study site as accurately as possible. For example, the spatial map of pH for the same area the present study was conducted six years prior reported very similar mean values (Fig. S4 from ¹).

We do not know how stable the seeping is prior to this. However, the seep sites are known as “Illi Illi Bua Bua” in the local language, meaning “blowing bubbles.” Local communities report the seeps have been active for at least three generations as reported previously ¹.

	We have added some additional information to the method and discussion sections acknowledging this uncertainty. Lines 332: “Seeping intensity is characteristically variable over short time-frames (minutes to hours), but mean CO₂ levels have remained relatively constant over six years of sampling (e.g. Table S2 and Fig. S3 from ¹ and the present study).” And line 305: “These CO₂ seep sites are not perfect representations of future coral reefs. They are small in size and lack the co-occurring elevated temperature stress expected under climate change. Seep CO₂ levels are also characteristically variable, especially within areas with low mean pH ¹ (Fig. S1), and effects of this variability are largely unknown for most coral reef organisms ². Hence extrapolating the results seen here to the future of coral reefs globally will attract some uncertainty. However, studies in situ at CO₂ seeps have advantages over laboratory studies (e.g. organisms grow under natural conditions of ecological interactions and have life-time acclimation) and hence provide unique insights into the effects of ongoing ocean acidification on marine communities.”
2)The benthic sampling that was conducted at each site needs some justification. Benthic composition (percent cover, diversity, etc.) at each site was quantified primarily from photographs of 4, 1-m² quadrats at each site. This strikes me as a fairly low sample size for coral reefs which are notably for their patchiness. Furthermore, these ecological data were extracted from the photographs by overlaying 28 randomly-located dots to	The seep site contains mosaic of gas streams, so carbon chemistry changes occur over short distances. The four quadrats per measurement station were placed in close proximity to the float in the middle so that the carbon chemistry measurement at the float would best represent the reef area occupied by the quadrats. Increasing the number of quadrats or the area surveyed per station would have decreased the representativeness of the pH and total alkalinity measurements. Our study design instead opted for an increased number of stations with a smaller area. We have updated the text to reflect this (see below). We disagree that 28 points per photograph is a small number for human annotated imagery. For example, the Australian Institute of Marine Science’s Long Term Monitoring Program conducts benthic

determine taxa abundances. This too is a modest number of sampling points per photograph for such a diverse community. Often 100s of points are used per photograph and many coral reef studies.	surveys of the GBR by identifying the benthos under five points per image, with 40 images per transect (see SOP10). Here we purposefully increased the number of points per image annotated per photograph to capture smaller and rare taxa. Modern techniques for image annotation with hundreds of points per image rely on image recognition software and are not yet sophisticated enough for the taxonomic resolution required for this study. For example, the prominent benthic image platform Reef Cloud can annotate hundreds of points per photograph, but to roughly 80% accuracy at the phyla level. Line 368: “Quadrats were placed immediately around the sub-surface floats, to ensure the measured carbon chemistry parameters from the floats were representative of the quadrat area.” We have also now added the number of annotated points from the imagery to the methods section (4004 points, Line 376).
3)The sampling of macroalgae (abundance and biomass) was conducted in 2, 0.25-m² sub-quadrats within only 2 of the 4 larger quadrats at each site. This too is a pretty small sample size for a diverse community. I can appreciate the need to allocate field efforts to various tasks, but under-sampling can have insidious effects on results. So I think some justification would be useful.	The biomass sampling was limited to the two sub-quadrats due to time constraints in the field. Each sampling dive took approximately one hour per station. We have included a description in the methods to justify this. Lines 386: “All macroalgae occurring within one quarter (i.e. 0.25 m²) of two of the quadrats per station were hand-collected by divers using a scraping tool. Larger areas were not sampled due to time constraints in the field.”
4)It probably is not a big deal, but the macroalgae were initially sun-dried in the field and then dried in an oven later at 45 degrees C. Why so low? Typically algae are dried at 60-80 degrees C to quantify dry mass.	This is a typo. The algae were dried at 60 °C in the lab. The manuscript text has been corrected.

5)I don't intend for the above comments to result in "death by a thousand cuts", but I do think that these issues need to be addressed in order for the reader to truly appreciate the importance of the effects that are presented.	We thank the reviewer for their constructive feedback. It has undoubtedly improved our manuscript.
--	--

Reviewer 3

Comment	Response
No actionable comments.	We thank the Reviewer for the positive comments.

References

1. Fabricius, K. E. *et al.* Losers and winners in coral reefs acclimatized to elevated carbon dioxide concentrations. *Nat Clim Chang* **1**, 165–169 (2011).
2. Rivest, E. B., Comeau, S. & Cornwall, C. E. The role of natural variability in shaping the response of coral reef organisms to climate change. *Curr Clim Change Rep* **3**, 271–281 (2017).
3. O'Brien, P. A. *et al.* Elevated CO₂ has little influence on the bacterial communities associated with the pH-tolerant coral, massive *Porites* spp. *Front Microbiol* **9**, (2018).